# An incoherent feedforward loop facilitates adaptive tuning of gene expression

Jungeui Hong[1,2], Nathan Brandt[1], Farah Abdul-Rahman[1], Ally Yang[3], Tim Hughes[3], David Gresham[1]*

[1]Department of Biology, Center for Genomics and Systems Biology,  New York University, New York, United States; [2]Memorial Sloan Kettering Cancer Center, New York, United States; [3]Banting and Best Department of Medical Research, Donnelly Centre, University of Toronto, Toronto, Canada

**Abstract** We studied adaptive evolution of gene expression using long-term experimental evolution of *Saccharomyces cerevisiae* in ammonium-limited chemostats. We found repeated selection for non-synonymous variation in the DNA binding domain of the transcriptional activator, GAT1, which functions with the repressor, DAL80 in an incoherent type-1 feedforward loop (I1-FFL) to control expression of the high affinity ammonium transporter gene, MEP2. Missense mutations in the DNA binding domain of GAT1 reduce its binding to the GATAA consensus sequence. However, we show experimentally, and using mathematical modeling, that decreases in GAT1 binding result in increased expression of MEP2 as a consequence of properties of I1-FFLs. Our results show that I1-FFLs, one of the most commonly occurring network motifs in transcriptional networks, can facilitate adaptive tuning of gene expression through modulation of transcription factor binding affinities. Our findings highlight the importance of gene regulatory architectures in the evolution of gene expression.

DOI: https://doi.org/10.7554/eLife.32323.001

*For correspondence:
dgresham@nyu.edu

## Introduction

Gene expression evolution is a pervasive source of phenotypic diversity between and within species (*Jacob and Monod, 1961*; *Jacob, 1977*; *Nocedal and Johnson, 2015*). Genetic variation causing evolutionary changes in gene expression impacts either the regulatory elements of a gene (*cis*-regulatory) or the factors that control its expression (*trans*-regulatory). The relative importance of these two classes of variation across evolutionary timescales is the source of long-standing debate (*Wittkopp et al., 2004*; *Hoekstra and Coyne, 2007*; *Stern and Orgogozo, 2008*; *Soskine and Tawfik, 2010*). A variety of approaches have been developed for identifying the genetic basis of intraspecific gene expression variation (*Brem et al., 2002*; *Yvert et al., 2003*) and defining the landscape of mutational changes that affect gene expression (*Metzger et al., 2016*). Comparative genomics using extant organisms is typically used for inferring the evolutionary processes and outcomes that underlie the evolution of gene expression (*Wittkopp, 2010*). However, these approaches face the challenge of distinguishing neutral from adaptive variation and cannot provide insights into the dynamics of gene expression evolution in evolving populations.

The regulation of gene expression typically comprises multiple regulatory factors with both activating and repressive functions that coordinately control the dynamics and output of gene transcription. Regulatory networks controlling gene expression are composed of network motifs, recurrently occurring patterns of interaction between regulatory factors (*Alon, 2007*). Studies of global regulatory networks in *Escherichia coli* (*Shen-Orr et al., 2002*) and *Saccharomyces cerevisiae* (*Lee et al.,*

*2002*) identified three component feed-forward loops (FFL) as one of the most commonly occurring network motifs. A FFL comprises a transcription factor that regulates a second transcription factor. Both transcription factors bind the regulatory region of a third target gene and jointly control its expression. As transcription factors can be either activators or repressors, there are eight possible configurations of three component FFLs (*Mangan and Alon, 2003*). Of these variants, incoherent type-1 feed-forward loops (I1-FFL) are one of the most common, occurring hundreds of times in bacteria and yeast gene regulatory networks (*Lee et al., 2002*; *Mangan and Alon, 2003*; *Milo et al., 2002*). These network motifs are termed 'incoherent' as the upstream regulator directly activates the target gene and indirectly represses it by activating a repressor of the target gene. I1-FFL function to speed the response time of gene expression (*Mangan and Alon, 2003*) and for particular parameter values, detect relative changes in input signals rather than absolute levels of inputs (*Goentoro et al., 2009*). 1I-FFLs are found throughout gene regulatory networks in multiple organisms. Although the dynamic properties and behavior of the I1-FFL have been extensively studied, the significance of this network in evolution is not well understood. In particular, how I1-FFLs constrain or facilitate evolutionary pathways remains unknown.

Long-term experimental evolution (LTEE) provides a means of studying evolutionary dynamics, processes and outcomes. Performing replicated LTEE in controlled conditions and assessing the fitness and phenotypes of evolved lineages and populations enables tests of the role of gene expression evolution in response to selection and the extent to which selection results in recurrent outcomes. LTEE in microorganisms provides evidence for the role of adaptive changes in gene expression in response to selection as parallel changes in gene expression have been observed in replicated populations of both *E. coli* (*Cooper et al., 2003*) and *S. cerevisiae* (*Ferea et al., 1999*; *Gresham et al., 2008*). However, proving that changes in gene expression are adaptive and identifying the underlying molecular bases of gene expression changes acquired during experimental evolution remains challenging.

Quantitative DNA sequencing enables the comprehensive identification of genomic variants, their frequencies and dynamics during adaptive evolution (*Gresham et al., 2008*; *Kao and Sherlock, 2008*; *Wenger et al., 2011*; *Hong and Gresham, 2014*). In LTEE performed in conditions of constant nutrient-limitation using chemostats, increased expression of nutrient transporter genes is a primary mode of adaptive evolution (*Gresham and Hong, 2015*). Copy number variants (CNVs), comprising gene amplifications that result in increased expression of transporter genes (*Gresham et al., 2008*; *Hong and Gresham, 2014*), are a recurrent class of adaptive alleles in chemostat LTEE. However, adaptive mutations in trans factors, such as transcription factors (TFs) are also frequently identified in LTEEs (*Kao and Sherlock, 2008*; *Hong and Gresham, 2014*). These variant alleles are primarily missense, frame-shift or nonsense mutations that are likely to confer strong effects on gene expression in evolved lineages (*Wray, 2007*; *Carroll, 2008*). However, how protein coding changes in transcriptional regulators identified during LTEE alter gene expression and how these changes in expression confer increased fitness remains unknown.

The regulation of nitrogen utilization in budding yeast represents one of the best understood gene regulatory networks in any system. Control of nitrogen catabolite repression (NCR) transcription is achieved by four transcription factors, that bind to the same GATAA consensus sequence (*Cooper, 2002*; *Magasanik and Kaiser, 2002*; *Hofman-Bang, 1999*). Two factors – GLN3 and GAT1 – have activating activity and two factors – DAL80 and GZF3 – have repressive activity. GLN3 is constitutively expressed and regulated post-translationally by TORC1-dependent phosphorylation that prevents its entry into the nucleus. In conditions in which TORC1 activity is low, dephosphorylation of GLN3 enables it to enter the nucleus where it activates expression of GAT1, DAL80 and GZF3 (*Figure 1—figure supplement 1*). The promoters of GAT1, DAL80 and GZF3 contain GATAA binding sites and thus they coordinately regulate expression of their targets and themselves. The NCR regulatory components comprise four interconnected I1-FFLs comprising either of the activators, GLN3 and GAT1, and either of the repressors DAL80 or GZF3, that function to regulate the expression dynamics and output of ~41 target genes (*Godard et al., 2007*) that encode products required for assimilating diverse nitrogen sources. NCR is an ideal system for studying the evolution of gene expression owing to the well-characterized properties of its four regulators and the small number of well-defined direct targets.

Here, we used LTEE in ammonium-limited chemostats to investigate the functional basis of adaptive increases in gene expression. Leveraging our understanding of selection in chemostats primarily

operating on nutrient transport (*Gresham and Hong, 2015*) and the detailed understanding of nitrogen-regulated gene expression in yeast (*Cooper, 2002*; *Magasanik and Kaiser, 2002*) we studied how adaptive variation in the trans regulatory factor, GAT1, mediates its effects on fitness by altering gene expression. We find that the beneficial effects of recurrently selected missense mutations in GAT1 are a result of its effect on increasing expression of the high affinity ammonium transporter gene, *MEP2*. Surprisingly, we find that these effects are the result of decreased affinity for the GAT1 consensus binding site that differentially impact binding at the promoters of the repressor *DAL80* and their common transcriptional target, *MEP2*. Using functional assays and mathematical modeling we find that adaptive evolution acts to increase the transcriptional output from the I1-FFL comprising GAT1, DAL80 and MEP2 by decreasing transcription factor affinities. Thus, in addition to their important dynamic behavior, I1-FFL possess properties that may be exploited by evolution to tune their output thereby increasing organismal fitness.

## Results

### De novo missense mutations in *GAT1* are positively selected in ammonium-limited chemostats

Previously, we identified repeated selection of independent missense mutations in the DNA binding domain of GAT1 within a single LTEE population maintained in ammonium-limited chemostats (*Hong and Gresham, 2014*). In this study, we aimed to determine whether adaptive mutations in *GAT1* result in alteration of its regulatory activities and, if so, how these changes impact fitness. First, we tested the repeatability of adaptive *GAT1* mutations in initially clonal populations maintained in ammonium-limited environments by performing LTEE in triplicate using asexually reproducing populations in ammonium-limited chemostats (*Figure 1a*). We found that evolution is highly parallel both at the phenotypic and genotypic levels over 250 generations. We observed significant increases in population level fitness with an overall deceleration in the rate of fitness improvement with time as seen in previous LTEEs (*Barrick et al., 2009*) (*Figure 1b*). Using whole genome, whole population sequencing of evolving populations, we identified recurrent selection for missense mutations in the DNA binding domain of *GAT1* during the early stages of adaptive evolution in each population. Consistent with positive selection for increased ammonium transport capacity in this environment, we also identified CNVs that include *MEP2*, which encodes a high-affinity ammonium transporter, at high frequencies following 250 generations of selection (*Figure 1c*).

Recently, lineage tracking during LTEE in large populations has shown that many independently derived beneficial alleles are present at low frequencies and unlikely to be detected using whole genome, whole population sequencing (*Levy et al., 2015*). Therefore, we investigated whether additional low frequency mutations of *GAT1* were present in evolving populations using targeted deep sequencing (*Figure 1d* and *Supplementary file 1A*; see Materials and methods). Each of the evolving populations contained multiple different *GAT1* mutations at frequencies of $10^{-2}$-$10^{-3}$. These GAT1 alleles reach their highest frequencies during the early stages of adaptive evolution before lineages containing *MEP2* CNVs ultimately rise to high frequency in each population. Importantly, all identified GAT1 mutations are missense mutations in its DNA binding domain (*Figure 1e*), which we find is under strong purifying selection in natural environments (*Figure 1f*).

### *GAT1* mutations increase fitness in ammonium-limited chemostats in a MEP2 dependent manner

To study the functional basis of adaptive *GAT1* mutations, we isolated clones with three individual alleles from distinct evolved lineages – *gat1-1* (W321L), *gat1-2* (C331Y), and *gat1-3* (R345G) – (*Figure 2a*) using backcrossing and allele specific PCR genotyping (*Hong and Gresham, 2014*). Using competition assays (see Materials and methods), we determined the fitness of the evolved lineages that contain *GAT1* mutations plus an additional 3–4 variants in other genes acquired during LTEE. These variants include two *MEP2* alleles that we previously identified as reversions of a lab acquired mutation in the transmembrane domain of MEP2 (*Hong and Gresham, 2014*). We also quantified the fitness of the *GAT1* mutations alone, in different nitrogen-limiting conditions – ammonium, glutamine, proline and urea – using chemostats and rich media (YPD) batch condition. All strains containing *GAT1* missense mutations are significantly increased in fitness in ammonium-

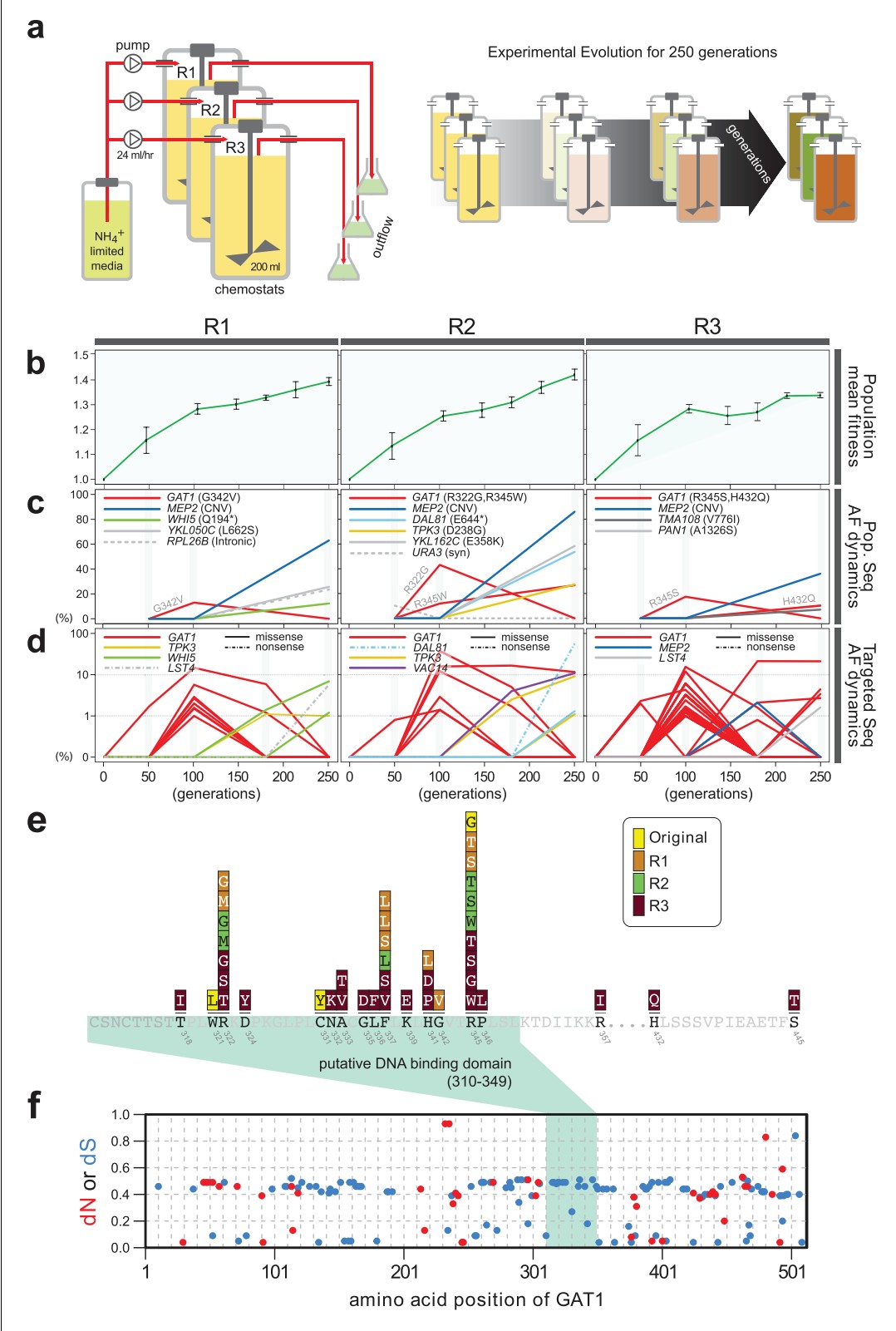

**Figure 1.** Repeated LTEE in ammonium-limited chemostats selects for adaptive mutations in *GAT1*. (a) LTEE design. Three chemostat vessels w ere founded with a clonal ancestral population and maintained under constant dilution with defined media containing 0.4 mM (NH 4) 2 SO 4 , (0.8mM nitrogen). Chemostats were maintained in continuous mode for 250 generations (~ 3 months) and samples obtained every 10-20 generations for archiving. (b) Dynamics of population fitness. The rate of fitness improvement decelerates over time for three independently replicated LTEE (R1, R2

*Figure 1 continued on next page*

*Figure 1 continued*

and R3). Error bar are 95% CI of linear regression analysis of competition assays, which comprised six time points each. (c) Allele dynamics in parallel LTEE. Whole population while genome sequencing was performed on samples from the three LTEEs at 50, 100 and 250 generations to identify high frequency mutations (> ~10%) using an Illumina HiSeq2500 in 2 x 50 bp paired end mode with an average read depth of ~50X. The frequency of the MEP2 amplification was defined as the proportion of clones bearing more than 2 copies of MEP2 among 96 randomly selected clones. GAT1 (red lines) variants are a primary target in the earliest generations of selection but, in some LTEE, are ultimately replaced by other alleles including CNVs that encompass MEP2 (blue lines) and mutations in genes that control cell cycle and growth. x- and y-axes represent time in generation and AF (allele frequency) in %, respectively. (d) Dynamics of minor frequency mutations. 12 genes that have previously been identified as adaptive targets of selection in different nitrogen-limited environments (see method) were subjected to targeted deep sequencing. Multiple missense mutations in GAT1 are present simultaneously in each population and compete with each other during the earliest generations. Discrepancies in the estimated AFs of the same mutations identified in *Figure 1c* are likely due to noise in the sequencing assay. x- and y-axes represent time in generation and AF (allele frequency) in %, respectively (e) Mutational landscape of adaptive GAT1 mutations. All mutations in GAT1 with population AF greater than 1% are shown. The DNA binding domain of GAT1 is a mutational hotspot in all ammonium-limited LTEE. Multiple missense (protein coding alteration) mutations were observed but no nonsense or frame-shift mutations were detected. (f), The DNA binding domain of GAT1 is under purifying selection in the wild. dN and dS values (see method for their definition) for GAT1 at each amino acid position were calculated using SNAP v2.1.1 (http://www.hiv.lanl.gov/content/sequence/SNAP/SNAP.html) *Korber, 2000* using sequences from 42 different wild yeast strains. Unlike LTEE in ammonium-limited chemostats, the GAT1 DNA binding domain is under purifying selection implying that non-synonymous mutations are likely to be detrimental in dynamic environments.

DOI: https://doi.org/10.7554/eLife.32323.002

The following figure supplement is available for figure 1:

**Figure supplement 1.** Model of NCR regulon.

DOI: https://doi.org/10.7554/eLife.32323.003

limited chemostats, but exhibit decreased fitness in all other nitrogen-limited conditions consistent with antagonistic pleiotropy (*Figure 2b*). By contrast, a *GAT1* knock out (KO) strain exhibits a fitness defect in both ammonium and proline limited chemostats. *GAT1* mutations are nearly neutral in YPD batch media conditions in which *GAT1*, and the entire NCR regulon, is transcriptionally repressed. Collectively, these results suggest that missense mutations in the DNA binding domain of GAT1 acquired during LTEE alter its function rather than rendering it non-functional.

Increased gene expression of nutrient transporters through gene amplification is a prevalent means of increasing fitness in nutrient-limited chemostats (*Hong and Gresham, 2014*). We asked whether the effects of beneficial *GAT1* mutations are mediated by their impact on *MEP2* regulation and therefore represent an alternate class of adaptive mutations with the same functional effect of increased transporter expression. To test the importance of *MEP2* for the increased fitness associated with *GAT1* variants we deleted either the ORF or the promoter region (1 kb upstream of the ORF) of *MEP2* in the background of the *gat1-1* and *gat1-3* alleles and tested their fitness in ammonium-limited chemostats. All strains had severely reduced fitness (*Figure 2—figure supplement 1a*) consistent with the beneficial effect of *GAT1* variants being dependent, at least in part, on *MEP2* expression. We also isolated evolved lineages with increased copy numbers of *MEP2* from each of the three LTEEs and quantified *MEP2* copy number, expression level and relative fitness in ammonium-limited conditions (*Figure 2—figure supplement 1b*). Increased copy number of *MEP2* correlates with increased mRNA expression and results in fitness increases (s > 1.3) greater than that conferred by GAT1 variants (*Figure 2b*). The detection of lineages with increased fitness and increased expression of *MEP2* resulting from CNVs is consistent with fitness increases in GAT1 variant lineages being attributable to their effect on *MEP2* expression.

### GAT1 variants result in increased expression of MEP2 but decreased expression of DAL80

We investigated the effect of adaptive missense mutations in *GAT1* on gene expression. We performed RNA-seq using three strains that contain one of each of the three adaptive *GAT1* mutations alone cultured in ammonium-limited chemostats. We compared expression profiles of *GAT1* adaptive mutations with those in the corresponding evolved lineages from which the mutations were isolated (*Hong and Gresham, 2014*). We found significant divergence of NCR gene expression in all tested strains (*Figure 3a* and *Supplementary file 1B*). *GAT1* mutations result in increased expression of genes encoding ammonium transporters (*MEP1*, *MEP2*, and *MEP3*) as well as transporters that import other nitrogen sources such as urea, allantoin and GABA. The evolved lineages, which

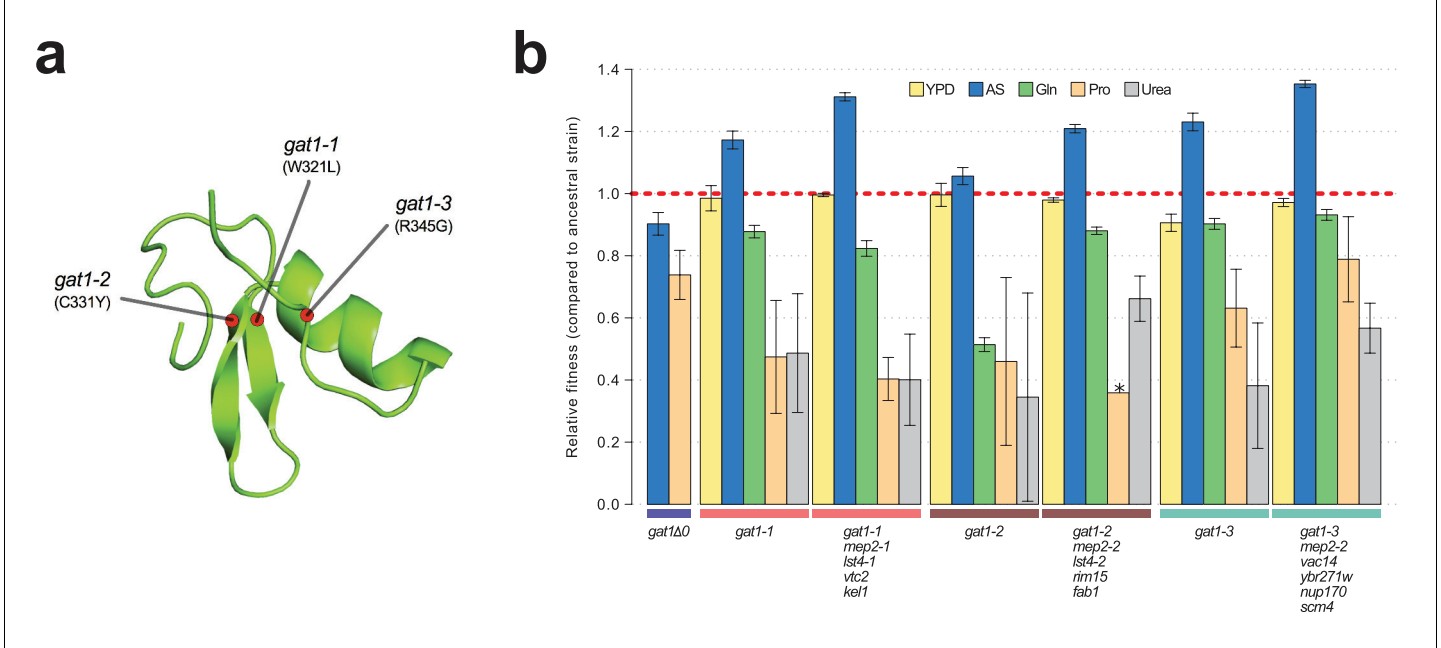

**Figure 2.** Adaptive *GAT1* mutations underlie increased fitness and confer antagonistic pleiotropy. (a) 3D structure model of GAT1 DNA binding domain. The 3D structure is based on available GATA factor DNA binding domain structures in the modbase database (https://modbase.compbio.ucsf. edu/). Adaptive amino acid changes selected for the subsequent analyses lie within, or in close proximity, to the DNA binding domain of GAT1. (b) Fitness effects of GAT1 mutations. GAT1 variants are beneficial in ammonium-limited chemostats and confer increased fitness in the lineages in which they occur. GAT1 variants confer a fitness cost in non-ammonium limited environments consistent with antagonistic pleiotropy. Error bars represent 95% CI of linear regression analysis of competitive fitness assays. '*' denotes the absence of a 95% CI estimate due to a small number of time points sampled in the fitness assay (<5). (YPD: YPD batch culture, AS: Ammonium sulfate-limited chemostat, Gln: Glutamine-limited chemostat, Pro: Proline-limited chemostat, Urea: Urea-limited chemostat; all nitrogen-limited media were normalized to 0.8mM nitrogen).

DOI: https://doi.org/10.7554/eLife.32323.004

The following figure supplement is available for figure 2:

**Figure supplement 1.** GAT1 variant fitness effects are due to its effect on MEP2.

DOI: https://doi.org/10.7554/eLife.32323.005

---

contain *GAT1* mutations as well as additional mutations, show more 'fine-tuned' gene expression in which only ammonium permease-encoding genes are increased in expression while other NCR targets, which are likely to be irrelevant when ammonium is the only nitrogen source, are repressed. By contrast, *GAT1* mutations result in a strong reduction in *DAL80* expression. The expression of *GLN3* and *GZF3* is unchanged in the presence of *GAT1* variants (*Supplementary file 1B*) suggesting that the altered expression of *MEP2* and *DAL80* is attributable to changes in GAT1 binding affinity. Interestingly, GAT1 variants also result in increased expression of GAT1 itself.

To further quantify changes in transcriptional activity attributable to *GAT1* variants, we fused the promoter sequences of four targets of GAT1 (*GAP1*, *MEP2*, *GZF3* and *DAL80*) to the coding sequence of GFP in the backgrounds of the ancestral (WT), KO (*gat1Δ0*), and the three GAT1 variants (*gat1-1*, *gat1-2* and *gat1-3*) (*Figure 3—figure supplement 1a*). GFP expression levels measured using this assay were highly comparable to RNA-seq data (*Figure 3—figure supplement 1b*) and therefore a good proxy of transcriptional activation of each promoter in the background of the different *GAT1* variants (*Figure 3—figure supplement 1c*).

We found significant differences in transcriptional activities between *GAT1* variants at *MEP2* and *DAL80* promoters in nitrogen-limited (both ammonium and proline) conditions (*Figure 3b* and *Supplementary file 1C*). All adaptive *GAT1* mutations result in strongly reduced *DAL80* expression and increased *MEP2* expression compared to the wild type, while a *GAT1* KO strain showed slightly reduced *DAL80* expression and decreased *MEP2* expression. This result is consistent with our RNA-seq analysis and provides additional evidence that the beneficial missense mutations in the DNA binding domain of GAT1 do not render it non-functional. We also tested whether these mutations

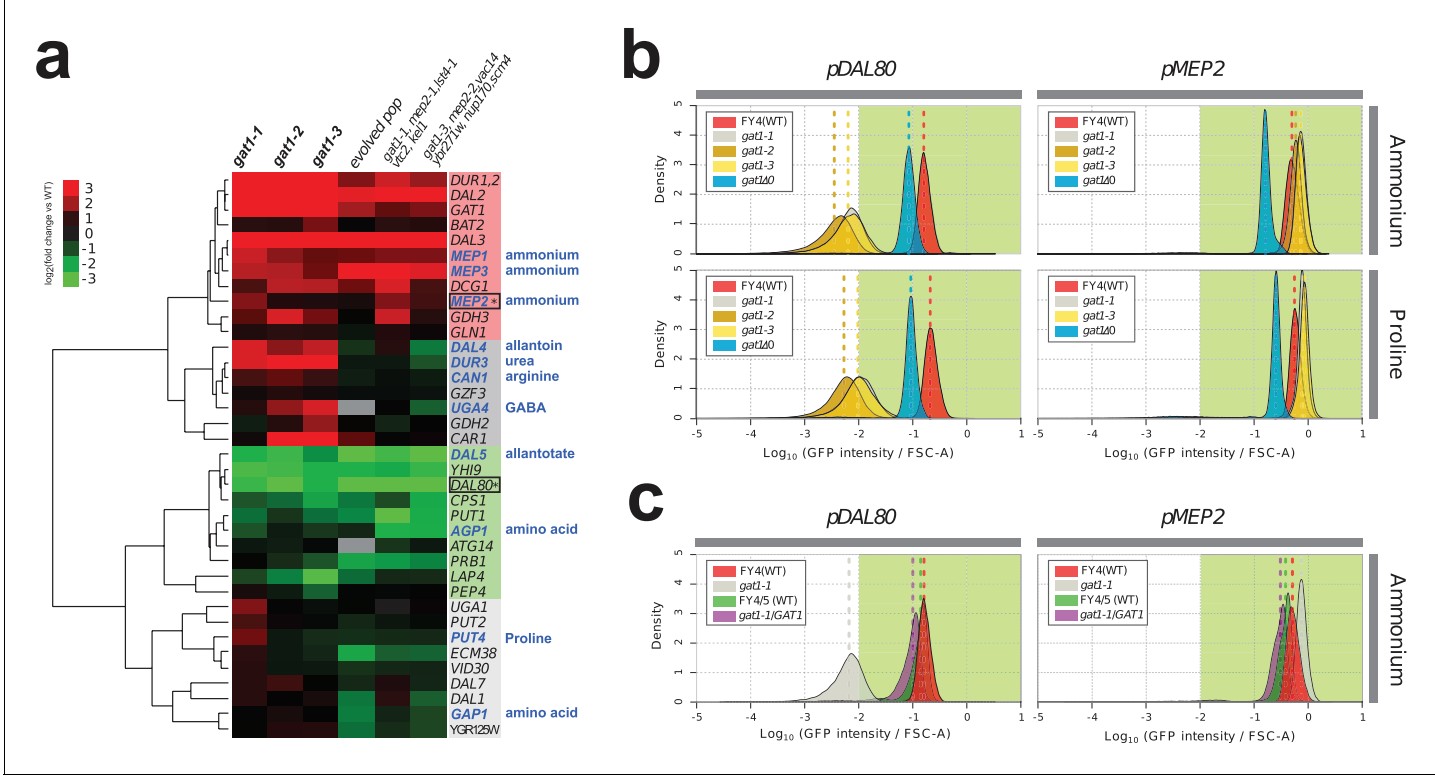

**Figure 3.** Adaptive *GAT1* variants exhibit differential effects on target gene expression. (a) Gene expression profile of NCR genes in different *GAT1* variant lineages. 'Blue' gene names indicate permease-encoding genes for different nitrogen sources. Clustered gene expression values relative to the ancestral strain (log₂ transformed fold change compared to the ancestor without statistical filtering) are shown for 41 experimentally confirmed NCR target genes. Expression data for the final evolved population and two lineages containing multiple mutations in addition to GAT1 variants from our previous report (*Hong and Gresham, 2014*) are also presented for reference. (b) Transcriptional reporter assay for *MEP2* and *DAL80*. We quantified GFP expression from the *MEP2* and *DAL80* promoters in the ancestral and *GAT1* variant backgrounds in different nitrogen-limited conditions. In nitrogen-limited chemostats containing different individual sources of nitrogen (ammonium, glutamine, proline and urea), all mutant strains (*gat1-1*, *gat1-2* and *gat1-3*) show the same gene expression pattern for *DAL80* (decreased expression) and *MEP2* (increased expression) compared to the ancestral genotype. A *GAT1* KO mutant results in decreased expression of both *DAL80* and *MEP2*. (c) GFP reporter assay for *MEP2* and *DAL80* promoters in heterozygous diploids. The *gat1-1*/*GAT1* genotype results in expression of *DAL80* or *MEP2* comparable to the ancestral strain. The distribution and median expression (dashed line) for each genotype is shown.

DOI: https://doi.org/10.7554/eLife.32323.006

The following figure supplements are available for figure 3:

**Figure supplement 1.** Construction and validation of transcriptional reporters.

DOI: https://doi.org/10.7554/eLife.32323.007

**Figure supplement 2.** GFP reporter assay for *MEP2* and *DAL80* in the background of *GLN3* KO.

DOI: https://doi.org/10.7554/eLife.32323.008

are dominant or recessive using diploid strains heterozygous for adaptive *GAT1* mutations (e.g. *gat1-1*/*GAT1*) (*Figure 3c*). *DAL80* and *MEP2* promoter activities in these heterozygotes were identical to the haploid ancestral strain (FY4) containing a wildtype *GAT1* allele indicating that the missense mutations of *GAT1* are recessive and therefore unlikely to have gain-of-function or dominant negative effects. The maintenance, but reduced, expression of *DAL80* and *MEP2* in the *GAT1* KO background (*Figure 3b*) is due to the activity of GLN3 as a *gat1 gln3* double KO results in no expression from the *DAL80* and *MEP2* promoters (*Figure 3—figure supplement 2*). *GLN3* is required for expression of *MEP2* in the presence of *GAT1* variants (*Figure 3—figure supplement 2*), likely due to its role in activating the entire NCR regulon, but in these strains the expression of *GLN3* is not increased and therefore does not contribute to the increased steady-state expression of *MEP2*.

## GAT1 mutations decrease affinity for its consensus binding site in both MEP2 and DAL80 promoters

To test whether adaptive *GAT1* mutations have acquired new binding specificities we used protein binding microarrays (PBMs) to assay the DNA binding domains of adaptive *GAT1* alleles (*gat1-1* and *gat1-3*) as well as the ancestral *GAT1* allele. Whereas the ancestral GAT1 DNA binding domain shows clear evidence of specificity for the GATAA consensus sequence, adaptive GAT1 alleles failed to show evidence for any significant binding specificity using PBMs (*Supplementary file 2A*) indicating that their functional and fitness effects are not exerted by the acquisition of new DNA binding specificities.

We quantified alterations in the affinity of the adaptive GAT1 alleles for their target DNA sequences using electrophoretic mobility shift assays (EMSAs) (*Figure 4a*, *Supplementary file 2B* and S6). We studied the binding of adaptive GAT1 TF to the promoter sequences of *MEP2* and *DAL80*, which showed discrepant effects on gene expression in the presence of adaptive *GAT1* alleles: increased expression in the case of *MEP2* and decreased expression in the case of *DAL80* (*Figure 3a* and *Figure 3b*). *MEP2* contains two distinct GATAA sequence motifs in its promoter region whereas *DAL80* possesses only one. Adaptive *GAT1* variants showed significantly decreased binding affinity to all target motifs (*Figure 4b*). Binding kinetics calculated based on a two parameter Michaelis-Menten model (*Stormo and Zhao, 2010*) show that adaptive *GAT1* mutations have a more detrimental effect on binding to the *DAL80* promoter GATAA site compared with the two *MEP2* promoter binding sites likely reflecting differences in motif strength (*Figure 4c* and *Supplementary file 1D*). This suggests that the adaptive *GAT1* alleles encode functional TFs that maintain sequence specificity for GATAA binding sites, but exhibit a quantitative decrease in affinity for their target sites compared to the ancestral allele. Whereas reduced affinity of GAT1 variants for the comparatively weak promoter of *DAL80* results in its reduced expression, expression from the comparatively strong *MEP2* promoter is increased in the presence of GAT1 variants despite a reduction in affinity for GATAA sites in the *MEP2* promoter. Consistent with a differential effect of reduced GAT1 binding affinity on different promoters, we identified a significant positive correlation between gene expression levels of NCR genes and the estimated affinity of each promoter for GAT1 (*Figure 4d*).

Interestingly, *GAT1-1* (W321L) shows a much stronger reduction in DNA binding affinity compared to *GAT1-3* (R345G) for both tested promoter sequences (*Figure 4b and c*). Two pieces of evidence suggest this difference may have important functional effects. First, we find that the residue (R345) altered in the *GAT1-3* allele is the most frequently mutated site in multiple LTEEs whereas only a single variant was identified at the site mutated in the *GAT1-1* allele (W321) (*Figure 1e*). Second, the *GAT1-3* variant results in higher relative fitness compared to the *GAT1-1* variant (*Figure 2b*). These observations suggest that intermediate reductions in GAT1 binding affinity may be more beneficial than strongly reduced binding. Consistent with this claim, complete loss of GAT1 results in decreased MEP2 expression (*Figure 3b*) and is deleterious (*Figure 2b*).

## Decreased TF affinities can result in increased expression from an I1-FFL

Using functional assays, we found that the impact of GAT1 variants on promoter binding affinities differs between the *DAL80* and *MEP2* promoters. As DAL80 is a negative regulator of *MEP2*, a quantitative decrease in DAL80 expression is expected to result in increased expression from the *MEP2* promoter. However, GAT1 variants also reduce affinity for the *MEP2* promoter sequence potentially decreasing its direct activation. Thus, while the overall topology of the GAT1-DAL80-MEP2 I1-FFL is maintained in evolved lineages, the increase in *MEP2* expression, and likely other targets with strong promoters (including *GAT1* itself), must be mediated by maintenance of direct binding at the *MEP2* promoter and an indirect effect through reduced activation of the repressor, *DAL80* (*Figure 4e*). Using a mathematical model of I1-FFLs we confirmed that a net increase in *MEP2* expression results when decreased activation at the *DAL80* promoter is the only functional effect of *GAT1* variation (*Figure 4f*). However, using this model we also find that a GAT1 variant that results in a decrease in direct activation of the *MEP2* promoter will result in increased expression of *MEP2* when activation of the *DAL80* promoter is also decreased. Increased *MEP2* expression, despite decreased activation of the *MEP2* promoter, is enhanced by a greater relative decrease in *DAL80* activation. Thus, our model of a I1-FFL is consistent with the finding that decreased binding affinity of GAT1, leading to

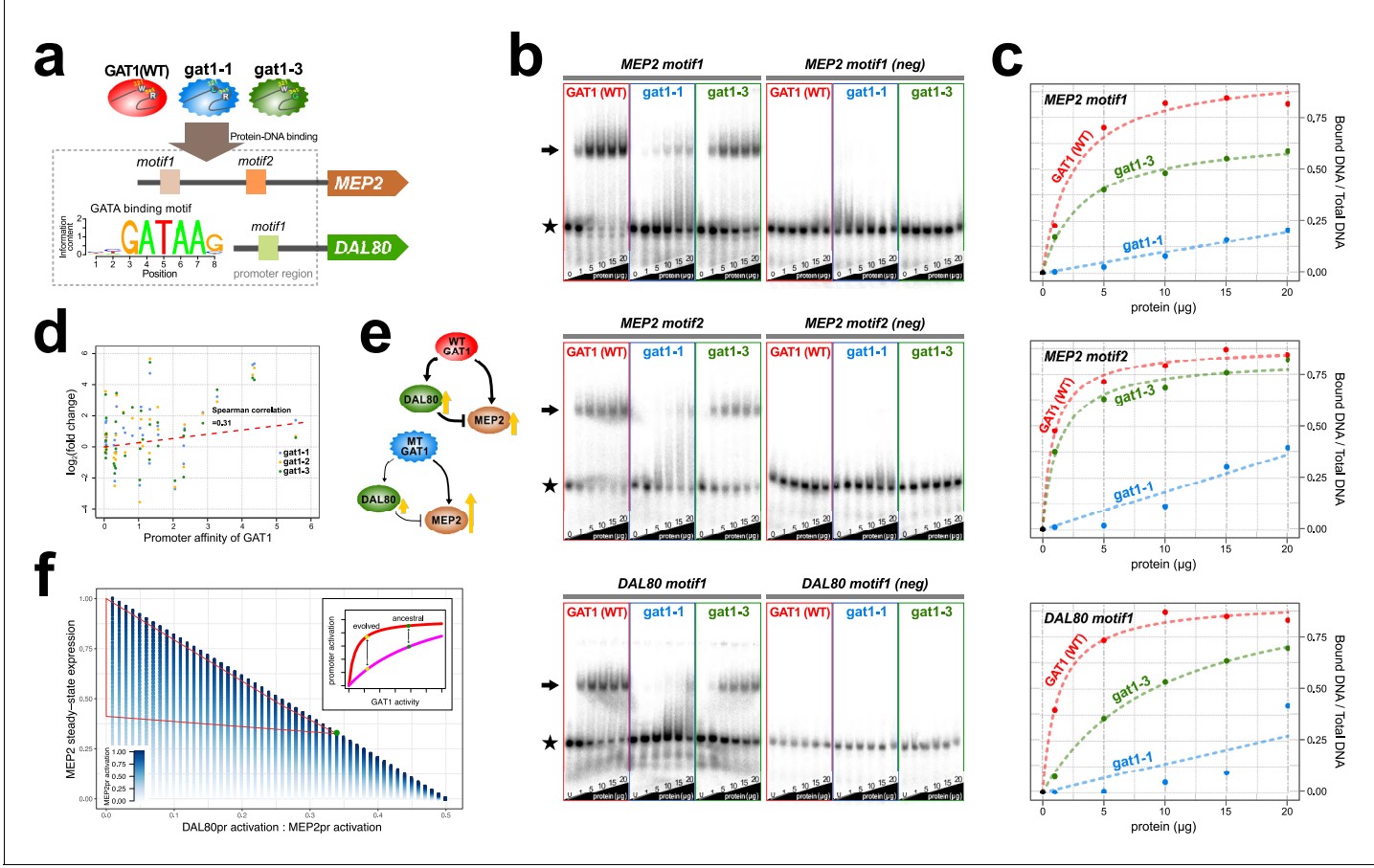

**Figure 4.** Adaptive *GAT1* variants alter DNA binding affinities in a promoter-specific manner. (a) Design of protein-DNA binding assays. We purified a 151-amino acid long protein fragments of *GAT1* containing the DNA binding domain (DBD) (positions 310–349) for the ancestral *GAT1* and two adaptive mutations – *gat1-1* (W321L) and *gat1-3* (R345G). DNA sequences containing the consensus sequence in *MEP2* and *DAL80* promoters were determined from the MotifDb collection in Bioconductor. MEP2 motif 1 contains three copies of GATAA, MEP2 motif 2 contains two copies of GATAA and DAL80 motif 1 contains one copy of GATAA. (b) Electrophoretic mobility shift assay (EMSA) analysis of *GAT1* variants. Increasing quantities of the GAT1 DBD protein fragment from 0 to 20 μg were added to $^{32}$P-labeled dsDNA. Arrows and stars indicate protein-bound DNA and unbound DNA respectively. Negative motif sequences (neg) were prepared by randomizing bases only at their GATAA binding motif sequences (see *Supplementary file 2B*). (c) Proportion of bound DNA for different amounts of protein. The two GAT1 variants showed significantly decreased binding affinity to all three GATAA motifs. Dotted lines are fits to a two-parameter Michaelis-Menten model. Only ancestral *GAT1* and *gat1-3* (R345G) showed significant estimates of two parameters ($Fr_{max}$ and $K_x$) (p-value<0.02). Insignificant estimates for *gat1-1* (W321L) may be due to the small number of data points (*Supplementary file 2C*). (d) Correlation between GAT1 promoter affinity and expression changes for NCR target genes. The promoter affinity for GAT1 (*Lee and Bussemaker, 2010*) is positively correlated (Spearman correlation = 0.31; p-value=0.02) with the alteration in expression of NCR targets in strains containing adaptive *GAT1* mutations (log$_2$ transformed fold change of gene expression of NCR genes in *GAT1* variant strains compared to the ancestor strain). (e) Model for altered output from the NCR regulatory network by adaptive *GAT1* mutations. *GAT1* mutations reduce affinity for the promoter of *MEP2*, but have an even greater reduction in affinity for the promoter of the repressor, *DAL80*. (f) Mathematical modeling of GAT1 affinity for *MEP2* and *DAL80* promoters on steady-state *MEP2* expression. The steady state expression level of *MEP2* depends on both the absolute activation of the *MEP2* promoter (indicated with increasing levels of blue) and *DAL80* promoter and the ratio of their levels of activation (x-axis). *MEP2* expression relative to an ancestral state (green point) is increased by adaptive GAT1 variants that result in a quantitative decrease in direct activation of *MEP2* and a proportionally greater decrease in direct activation of *DAL80* (parameter space resulting in increased *MEP2* expression outlined by red triangle). As shown in the inset, an ancestral state in which *MEP2* promoter activation (red) is greater than *DAL80* promoter activation (pink) can evolve to a state in which both promoters are decreased in activation, but the decrease is more pronounced at the *DAL80* promoter.

DOI: https://doi.org/10.7554/eLife.32323.009

The following figure supplements are available for figure 4:

**Figure supplement 1.** Increasing GAT1 expression results in increased MEP2 expression.
DOI: https://doi.org/10.7554/eLife.32323.010
**Figure supplement 2.** Purification of GST-tagged DNA binding domains of GAT1 variants.
DOI: https://doi.org/10.7554/eLife.32323.011

decreased activation of both the *DAL80* and *MEP2* promoters (*Figure 4b* and *Figure 4c*) can simultaneously result in decreased expression of the repressor, *DAL80* and increased expression of the target gene, *MEP2* (*Figure 3a* and *Figure 3b*). Using RNA-seq, we also find that the expression of *GAT1* is increased as a result of GAT1 variants (*Figure 3a*). As *GAT1* activates its own expression and is also repressed by DAL80, increased expression of *GAT1* is likely due to the same phenomenon that results in increased *MEP2* expression. The increase in *GAT1* expression likely also contributes to increased *MEP2* expression, as modeling an increase in *GAT1* expression results in increased *MEP2* expression (*Figure 4—figure supplement 1*).

## Discussion

Here, we describe the evolution of a gene regulatory network in real time and dissection of the functional basis of adaptive changes in transcription factor binding affinity. We find that missense mutations in *GAT1* are under strong positive selection in populations evolving in a constant ammonium-limited environment. The functional effects of adaptive *GAT1* mutations are the result of increasing the output from an I1-FFL, one of the most frequently occurring network motifs in transcriptional networks, by reducing the binding affinity of the activator, *GAT1*. The functional effect of adaptive GAT1 variation is mediated by increased expression of a high affinity ammonium transporter that results in increased fitness in the ammonium-limited environment.

### Increased sequencing resolution identifies multiple adaptive alleles in evolving populations

To date, genetic diversity and the dynamics of evolving lineages during LTEE has been largely explored using whole population whole genome sequencing (*Hong and Gresham, 2014*; *Lang et al., 2013*; *Kvitek et al., 2013*). However, the development of high-resolution lineage tracking using random molecular barcodes revealed that a multitude of lineages with increased fitness compete during the early stages of adaptive evolution, but are subsequently outcompeted by a small number of lineages carrying high-fitness mutations that occur early in the populations' histories (*Levy et al., 2015*). Consistent with this observation, we find that targeted deep sequencing of evolving populations identified many *GAT1* mutations in evolving populations that are under positive selection, but never increase to population frequencies greater than 1%. These beneficial mutations are not detected by population-level sequencing. Thus, defining the complete spectrum of adaptive mutations in evolving populations requires methods for detecting low frequency mutations that are beneficial but do not persist in the population. This is currently most feasible using the approach we adapted in this study: LTEE in defined environments and ultra-deep sequencing of loci that are known, or suspected, to harbor adaptive variation.

### Transcription factor binding affinities and promoter strength determine the output of network motifs

The transcriptional output from a promoter is a function of the strength of the promoter, which depends on the identity and number of cis regulatory elements, and the affinity of a transcription factor for those elements. Promoters are bound by both activating and repressing transcription factors and the relative activities of these trans factors coordinately controls the amount of expression from a given promoter. Here, we have shown that decreasing the binding affinity of GAT1 for its target sequence increases expression output from the *MEP2* promoter as well as other GAT1 targets with strong promoters. We attribute the increase in *MEP2* expression to decreased affinity of GAT1 for its consensus sequence, which reduces transcriptional output from the *DAL80* promoter, a comparatively weak promoter. A reduction in the effective concentration of the repressor results in increased transcriptional output from the *MEP2* promoter. Surprisingly, we also observe reduced affinity of GAT1 variants for the *MEP2* promoter, which one would expect might reduce its transcriptional output. However, as illustrated by our model (*Figure 4f*), a proportionally greater decrease in *DAL80* expression can outweigh the effect of reduced activation of the *MEP2* promoter. GAT1 is auto-regulatory and we observe that GAT1 variants also result in increased *GAT1* expression, which further increases expression output from the *MEP2* promoter (*Figure 4—figure supplement 2*). Thus, quantitatively different effects of transcription factor variants on promoters of differing

strengths, in the context of gene regulatory architectures, can lead to counterintuitive impacts on gene expression.

## The long-term fate of *GAT1* mutations in evolving populations

In our repeated LTEE we detected multiple *GAT1* mutations during the first ~100 generations of selection in ammonium limitation. The preponderance of these mutations within and between populations suggests that remodeling the GAT1-DAL80-MEP2 I1-FFL is a frequently occurring repeatable mode of adaptive evolution during the earliest stages of adaptive evolution in ammonium-limited environments. However, adaptive *GAT1* alleles likely do not represent a fitness optimum in the ammonium-limited environment as, in contrast to our previous study (*Hong and Gresham, 2014*), we found that ultimately these alleles were outcompeted by lineages containing *MEP2* amplifications. *MEP2* amplifications have the same functional effect as adaptive *GAT1* mutations: they increase expression of *MEP2* enabling improved ammonium transport capabilities. However, the increased expression of other high affinity transporters in *GAT1* lineages that are unable to transport ammonium may confer a fitness cost that is not associated with *MEP2* amplification alleles. It should be noted that our method for detecting MEP2 amplifications has limited detection power until CNVs rise to high frequencies in the population. Thus, it is likely that *MEP2* amplification alleles co-exist with adaptive *GAT1* mutations during the early phases of evolution but are undetectable using our sampling and assay scheme.

   Our results provide insight into NCR regulation and suggest additional levels of regulatory complexity. First, it is notable that we do not detect *DAL80* loss of function alleles in our LTEE. Given the current model of NCR regulation, loss of the repressor DAL80 would be expected to result in increased expression of *MEP2* and other NCR regulated genes resulting in increased fitness. Similarly, we do not see selection for variation in the other NCR repressor, *GZF3*. Finally, *GLN3*, the upstream activator of NCR genes, which is itself not a target of NCR regulation does not appear to harbor any adaptive variation. Combining LTEE with targeted studies of functional and fitness effects of regulatory components of NCR may provide further insights into how these four GATA factors coordinately regulate the NCR regulon.

## Parallels with oncogenic transcription factors mutations in human tumors

Experimental evolution in eukaryotic microbes has many parallels with the molecular processes, and evolutionary dynamics, underlying tumorigenesis. The evolutionary hotspot in the DNA binding domain of *GAT1* identified in this study is reminiscent of hotspots in transcriptional regulators important in tumorigenesis including recurrent missense mutations in the DNA binding domain of TP53, a driver in the majority of human cancers (*Freed-Pastor and Prives, 2012*), and CTCF, a poly-zinc finger transcription factor regulating oncogenes and tumor suppressor genes (*Ong and Corces, 2014*). As with adaptive evolution, missense mutations in the DNA binding domain of these transcription factors result in altered binding affinities at the promoters of target genes resulting in aberrant transcriptional activation in various cancers (*Park et al., 1994*; *Inga et al., 2001*; *Filippova et al., 2002*). Understanding the gene regulatory architecture of transcription factors that play important roles in human cancers will undoubtedly contribute to understanding the functional effect of oncogenic variation.

## Conclusion

The study of evolutionary processes using LTEE enables testing the dynamics, outcome and functional basis of adaptive evolution. Selection in chemostats provides the benefit of defined selective conditions and testable hypotheses regarding the functional basis of increased fitness (*Gresham and Hong, 2015*; *Gresham and Dunham, 2014*). Our study highlights the importance of network motifs and incorporation of biophysical parameters in gene regulatory models in order to predict their functional output in the context of evolution. As I1-FFL are one of the most frequent motifs in transcriptional networks they may serve a general evolutionary role as facilitators of transcriptional tuning through modulation of transcriptional factor binding affinities. Further work is required to determine whether the properties of I1-FFL afford this motif unique evolutionary potential.

## Materials and methods

### Strains and media

The ancestral strain used for all LTEE is a prototrophic haploid derivative (FY4; MATa) of the S288c reference strain. To isolate individual *GAT1* mutations, we crossed each evolved lineage to an isogenic haploid strain of the opposite mating type (FY5; MATα) and, following sporulation, recovered meiotic products bearing different combinations of derived adaptive alleles. To genotype individual segregants, we used allele-specific PCR for which two allele specific forward primers and one common reverse primer were prepared. We also knocked out the entire *GAT1* locus by replacing it with nourseothricin resistance (NatR) marker using a standard high efficiency transformation protocol. We engineered GFP fused promoter constructs for four GAT1 targets (*GAP1*, *MEP2*, *DAL80* and *GZF3*) and integrated them at the *HO* locus using homology based transformation (*Figure 3—figure supplement 1*). For each construct, we fused GFP with 1 kb of sequence 5' and 1 kb of sequence 3' of the open reading frame (ORF) for each target gene, which encompasses regulatory elements including promoter and UTRs. We attempted to construct a GFP-based transcriptional reporter using the *GAT1* promoter but found that including as much as 2 kb of sequence 5' to the *GAT1* ORF failed to activate GFP expression in either ancestral or adaptive *GAT1* backgrounds (data not shown). We also knocked out the *MEP2* ORF and 1 kb of sequence upstream of the *MEP2* ORF in the background of the ancestral strain as well as two strains bearing single *GAT1* missense mutations (*gat1-1* and *gat1-3*) using the same transformation protocol. We constructed diploid strains that are heterozygous at the *GAT1* locus by mating the GFP reporter constructs in the ancestral background and three *GAT1* mutant backgrounds to an isogenic strain of the opposite mating type (FY5). All constructed strains were verified using Sanger sequencing. All medium conditions and recipes were identical with those used in our previous study (*Hong and Gresham, 2014*): all nitrogen-limited media contained 0.8 mM of nitrogen regardless of the molecular form.

### LTEEs

All conditions for replaying LTEEs in chemostats were identical to those described in our previous study (*Hong and Gresham, 2014*). The dilution rate (0.12 hr$^{-1}$) of the chemostat was checked every one or two days over the 250 generation (~3 months) duration of the LTEEs, and intermediate samples of each population were archived every 20 generations. We confirmed that the steady-state cell density of ~3 $\times$ 10$^7$ cells/mL was consistently maintained in the 200 mL cultures over the entire LTEE.

### Whole genome, whole population sequencing

We performed whole genome sequencing of entire populations at 50, 100 and 250 generations for all LTEE cultures. Sequencing library preparation was performed using the Illumina TruSeq library preparation protocol as previously described (*Hong and Gresham, 2014*). We used 2 $\times$ 50 bp paired end mode on a HiSeq 2500 and all fastq files were processed using bwa (*Li and Durbin, 2009*), samtools (*Li et al., 2009*), and SNVer (*Wei et al., 2011*) to generate a list of mutations with significant allele frequencies in each population. The average sequence read coverage was around 50x, resulting in a detection limit for SNPs around 10% (i.e. we required that each variant was present in at least five reads). We identified CNVs by normalizing read depth at each nucleotide position to the median read depth of the entire genome by processing BAM files using samtools and R. We identified CNVs that include *MEP2* in all three populations at 250 generations using read depth analysis. At each time point during the LTEE, we estimated the allele frequency of CNVs in the population by randomly selecting 96 clones and conducting qPCR to measure the copy number of *MEP2* for all clones. We found that most clones possess one or two copies of *MEP2* but some possesses up to eight or nine copies (data not shown). All clones containing two of more copies of *MEP2* were aggregated within each population for allele frequency estimates of CNVs.

### Targeted amplicon sequencing

We performed targeted deep sequencing for 12 loci: *GAT1*, *MEP2*, *LST4*, *VAC14*, *RIM15*, *YBR271W*, *RPL26B*, YKL050C, *WHI5*, *DAL81*, *TPK3* and YKL162C. These loci are either genes that (1) contain high frequency SNPs identified in population sequencing or (2) have been repeatedly identified as

containing acquired variants in prior LTEEs in nutrient-limited chemostats. For each locus, we amplified the entire coding sequence plus 1 kb upstream and downstream of the ORF using PCR. We then randomly fragmented PCR products using sonication to a size of ~250 bp and prepared DNA sequencing libraries using the identical protocol used for population sequencing. Amplicon libraries were sequenced using an Illumina MiSeq in a 2 × 250 bp mode such that the two paired reads overlapped and each insert was sequenced twice. Only overlapping regions were used for allele frequency estimation and thereby we were able to dramatically reduce the false positive SNP call rate. For SNP frequency estimation, the average read coverage from amplicon sequencing was ~50,000. Using SNVer, we determined minor allele frequency mutations for all variants present at a frequency of $\geq 1\%$.

## dN/dS test

We obtained 42 different GAT1 sequences from multiple yeast isolates that were reported in a previous study (*Liti et al., 2009*). These include 32 natural isolates collected from multiple geographical locations including Europe, Africa, America, Asia and Oceania, and 10 domesticated isolates from sake or wine fermentations. dN and dS represent the proportion of observed nonsynonymous substitutions among all potential nonsynonymous substitutions and the proportion of observed synonymous substitutions among all potential synonymous substitutions, respectively. Actual values for each amino acid position were calculated from SNAP v2.1.1 (http://www.hiv.lanl.gov/content/sequence/SNAP/ SNAP.html)(*Korber, 2000*).

## 3D structural analysis of the GAT1 DNA binding domain

The ModBase model of yeast GAT1 (UniProt P43574) from an NMR structure (PDB 4GAT) complexed with Zn and DNA, with a nearly identical fold to a high-resolution crystal structure (PDB 2VUS), is the best model available. The 3D image was visualized by Polyview-3D (http://polyview.cchmc.org/polyview3d.html).

## Competitive fitness assays

Competition assays were performed as in (*Hong and Gresham, 2014*). We used a mCitrine-labeled ancestral (FY4) strain (DGY500, see *Supplementary file 1E*) for all competition assays. For population-level fitness assay, we established cultures containing the ancestral mCitirine-labeled strain (DGY500) alongside chemostats vessels in which LTEE were performed and let them reach the steady state conditions at the same dilution rate. We then mixed a sample from the LTEE with the fluorescence labeled ancestor strain in ratio of 1:9. To determine population fitness, each competing culture was sampled at multiple (5 -10) time points over a total period of less than 20 generations. At least 10,000 cells were assayed using an Accuri flow cytometer to determine the relative abundance of each genotype, and relative fitness determined using linear regression of the natural logarithm of the ratio of the two genotypes against the number of elapsed generations. We tested the fitness of three evolved mutants, three segregants from backcrossing experiments that contained only *GAT1* mutations and one engineered *GAT1* KO strain in four different nitrogen-limited chemostat conditions – using either ammonium, glutamine, proline or urea – and in YPD rich medium using batch culturing and serial dilution. We also performed fitness assays for *MEP2* ORF and promoter deletion strains in the ancestral background, and in the presence of *gat1-1* and *gat1-3* alleles, and three evolved lineages harboring additional *MEP2* gene copy selected from the finally evolved populations, using the same procedures in ammonium-limited chemostats. For the competition assay in YPD rich condition, we sampled twice per day and back-diluted the culture (1/200) into a fresh medium every night for 3 or 4 days. All subsequent analysis steps for clonal fitness determination were the same as population fitness assays.

## RNA-seq

We performed RNA-seq analysis in strains containing one of four different *GAT1* alleles (ancestral *gat1*, *gat1-1*, *gat1-2*, *gat1-3*) and one of three strains different GFP-based transcriptional reporter for either the *GAP1*, *MEP2* or *DAL80* promoter (*Supplementary file 1E*). Each set of experiments using strains that contained the same *GAT1* allele, but different transcriptional reporters was treated as biological triplicates as these strains are identical except for the transcriptional reporter. In total,

we cultured 12 different chemostat vessels (3 GFP reporters x 4 different *GAT1* alleles) and harvested a sample of 10 mL from each culture at steady state using vacuum filtration and snap freezing in liquid nitrogen with subsequent storage at −80°C until further processing. RNA extractions were performed using an acid phenol-chloroform method. For mRNA enrichment, we used poly-A selection and the final yield of selected RNA molecules was around 10–50 ng in total. For cDNA synthesis, we used Superscript III kit (Invitrogen) and dNTPs mixtures for the first strand synthesis and *E. coli* DNA ligase and polymerase I (Invitrogen) for second strand synthesis with a mixture of dATP, dCTP, dGTP and dUTP. Then, we performed end-repair, A-tailing and adapter ligation based on the standard Illumina TruSeq library preparation protocol. All reaction cleanups between each step were performed using AMPure XP beads (Beckman Coulter, Inc). For directional sequencing of first strand cDNA only, we treated with UNG (Thermo) and amplified the ligated molecules using Phusion high fidelity DNA polymerase (NEB) using 12 or 15 of PCR cycles. Adapter dimers were removed by conducting two rounds of size selection using AMPure XP beads. We confirmed the expected size distribution of ligated molecules using a BioAnalyzer and then quantified library concentrations using qPCR and KAPA Library Quantification kits (KAPA Biosystems). All RNA-seq was performed using an Illumina HiSeq 2500 in a 2 × 50 bp paired end rapid-run mode. We used Tophat (*Trapnell et al., 2009*) to align sequencing reads to the *Saccharomyces cerevisiae* S288C reference genome, obtained from SGD on Feb 03, 2011. From the resulting RNA-seq count data, we used edgeR (*Robinson et al., 2010*) to determine significant differential expression (DE) for all genes compared to the ancestral strain grown in the same conditions (see the final $\log_2$ transformed fold change values in *Supplementary file 1B*). The pearson correlation between promoter affinity for GAT1 and gene expression fold change was calculated for NCR genes only and a p-value was determined using permutation testing by randomly selecting 10,000 gene sets of the same size from all ~6000 yeast genes.

## GFP reporter assays

All reporter strains were grown in the same chemostat conditions at a dilution rate of 0.12 h$^{-1}$. Samples were taken at steady state, sonicated, resuspended in PBS buffer, and analyzed using an Accuri flow cytometer. In addition, we tested GFP expression in minimal media batch cultures containing either ammonium, glutamine, or proline as well as YPD batch culture. All GFP intensity values were normalized by forward scatter (FSC-A) and $\log_{10}$ transformed.

## Protein binding microarrays

PBM methods were identical to those described in (*Lam et al., 2011*; *Weirauch et al., 2013*). The DNA binding domains of GAT1 (WT) and two mutants (*gat1-1* and *gat1-3)* flanked by 50 amino acids on each side (see protein sequences in *Supplementary file 2B*) were subcloned into a GST-tag conjugated vector pTH6838 (*Supplementary file 1F*). Each plasmid was analyzed in duplicate on two different arrays (HK and ME) with differing probe sequences. We used 150 ng of plasmid DNA in a 15 µl in vitro transcription/translation reaction using a PURExpress In Vitro Protein Synthesis Kit (New England BioLabs) supplemented with RNase inhibitor and 50 µM zinc acetate. Calculation of 8-mer Z- and E-scores was performed as previously described (*Berger et al., 2006*). Microarray data were processed by removing spots flagged as 'bad' or 'suspect', and employing spatial de-trending (using a 7 × 7 window centered on each spot) as in (*Weirauch et al., 2013*). Experiments were deemed successful if at least one 8-mer had an E-score >0.45 on both arrays, the complementary arrays produced highly correlated E- and Z-scores, and the complementary arrays yielded similar PWMs based on the PWM_align algorithm, which aligns the top ten 8-mers (based on E-scores), and tallies the frequency at each position to generate a PWM (*Weirauch et al., 2013*).

## Electrophoretic mobility shift assays (EMSA)

We cloned the DNA binding domains of GAT1 (WT) and two mutants (*gat1-1* and *gat1-3)* flanked by 50 amino acids on each side (see protein sequences in *Supplementary file 2B*) into a plasmid backbone (pGEX 4 T-2; see *Supplementary file 1F*) containing a GST-tag, and introduced them into competent *E. coli* strains. GST-tagged proteins (~43 kDa) were induced and purified using glutathione. Purified products were analyzed using SDS-PAGE and Coomassie blue staining (*Figure 4—figure supplement 1*). EMSAs were performed by incubating increasing amounts of purified protein (0,

1, 5, 10, 15, and 20 µg) with a constant amount of a radiolabeled dsDNA corresponding to the GATAA motif present in the *MEP2* and *DAL80* promoters (see oligonucleotide sequences in *Supplementary file 2C*). Following electrophoresis, the gel was exposed to a phosphoimager screen for 30 min and scanned using a Typhoon. The intensity of the bands was acquired using ImageJ and the fraction of the intensity of the bound oligo over the total intensity of all bands was calculated. These data were fit to a two parameter ($Fr_{max}$ and $K_x$) Michaelis-Menten model using the drm (Dose-Response Model) function from the drc (Dose-Response Curve) package in R as followings (see *Supplementary file 1D*):

$$Fr = Fr_{max} \times \frac{x}{x + K_x}$$

where $Fr$ = fraction of bound DNA over the total DNA,

$x$ = input protein dose (µg),

$Fr_{max}$ = upper limit of $Fr$,

$K_x = \frac{Fr_{max}}{2}$, the protein quantity at which binding is half-maximal (µg)

## Mathematical modeling

We modeled the behavior of the incoherent feedforward loop comprising GAT1, DAL80 and MEP2 to study the effect on steady-state expression of MEP2. We modeled expression using the coupled ordinary different equations:

dGAT1/dt = 0

dDAL80/dt = A·X·GAT1 – c·DAL80

dMEP2/dt = A·GAT1 – R·DAL80 – c·MEP2

where

A = strength of promoter activation by GAT1

R = strength of MEP2 promoter repression by DAL80

X = ratio of MEP2 and DAL80 promoter activation by GAT1

c = degradation rate constant

We solved the system of equations numerically to determine the steady-state levels using the deSolve() package in R (*Soetaert et al., 2010*). We systematically varied the values of *A* and *X* to determine the effect of these parameters on the steady-state expression of *MEP2*. We set the parameter values of R = 2 and c = 1 and solved the system of equations using initial values of *GAT1* = 1, *DAL80* = 0, *MEP2* = 0. Variation in the parameter values of R and c alters the specific steady-state values, but the overall trend is robust to variation in these parameters.

## Sequence data

All raw fastq files are available from the NCBI Sequence Read Archive with accession number SRP101365 for whole genome population sequencing of replayed LTEE cultures, SRP101367 for targeted deep sequencing of the same LTEE populations and SRP101370 for RNA-seq data.

## Acknowledgements

We thank the Genomic Core Facility at New York University Center for Genomics and Systems Biology for DNA sequencing services. We thank members of the Gresham lab and Harmen Bussemaker for helpful discussions. This work was funded by the National Institute of Health (R01GM107466) and the National Science Foundation (MCB1244219). This paper is subject to the NIH Public Access Policy.

## Additional information

### Funding

| Funder | Grant reference number | Author |
|---|---|---|
| National Science Foundation | MCB1244219 | David Gresham |
| National Institutes of Health | R01GM107466 | David Gresham |

The funders had no role in study design, data collection and interpretation, or the decision to submit the work for publication.

## Author contributions

Jungeui Hong, Validation, Investigation, Methodology, Writing—original draft; Nathan Brandt, Ally Yang, Investigation, Methodology; Farah Abdul-Rahman, Investigation; Tim Hughes, Supervision; David Gresham, Conceptualization, Formal analysis, Supervision, Funding acquisition, Investigation, Methodology, Writing—original draft, Project administration, Writing—review and editing

## Author ORCIDs

David Gresham (iD) http://orcid.org/0000-0002-4028-0364

## Decision letter and Author response

Decision letter https://doi.org/10.7554/eLife.32323.016
Author response https://doi.org/10.7554/eLife.32323.017

# Additional files

## Supplementary files

• Supplementary file 1. (A) SNPs identified in each LTEE. Variants identified in each LTEE at four different time points throughout the evolution experiments and present at an allele frequency of at least 1% are listed. (B) Global gene expression analysis. Mean expression values for each gene for three *GAT1* variants relative to the ancestral genotype as determined by RNAseq are presented as $\log_2$ values along with an estimate of statistical significance. Gene expression of lineages and the population studied in (*Hong and Gresham, 2014*) are shown for reference. (C) Quantification of promoter activity using transcriptional reporters. The median fluorescence for each transcriptional reporter in each of the four *GAT1* genotypes relative to the ancestral strain was determined in different conditions. (D) Quantification of transcription factors binding affinity. Parameters estimated from fitting a Michaelis-Menten function to EMSA data. Fr_max is the maximal of bound over total DNA. K_x is the concentration of protein required for half-maximal binding. Non-significant parameter values are in grey italics. (E) Strains used in this study. (F) Plasmids used in this study.
DOI: https://doi.org/10.7554/eLife.32323.012

• Supplementary file 2. (A) Determination of GAT1 variants binding specificities by Protein Binding Microarray. Each purified protein was assayed on two different microarrays containing different probe sequences (ME or HK). E-scores for the top 8-mer bound by the protein on each microarray were calculated. A PWM was generated for each protein variant based on an alignment of the top ten 8-mers. (B) GAT1 protein fragment sequences used in PBM and EMSA assays. The DNA binding domain is in blue. Flanking sequences included in the expressed protein fragment are in green. (C) Oligonucleotides used to generate duplex DNA for EMSAs. Bold and underlined sequences are GATA sequences bound by GAT1. The minimum match score was set as 0.8 using 'matchPWM()' function in 'Biostrings' library in R. Negative motifs differ from target motifs only at the underline GATA by replacement with random sequence.
DOI: https://doi.org/10.7554/eLife.32323.013

• Transparent reporting form
DOI: https://doi.org/10.7554/eLife.32323.014

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
