## [Decision Letter]

Thank you for submitting your article "An incoherent feedforward loop facilitates adaptive tuning of gene expression" for consideration by *eLife*. Your article has been reviewed by three peer reviewers, and the evaluation has been overseen by Naama Barkai as the Senior and Reviewing Editor. The following individual involved in review of your submission has agreed to reveal his identity: Kevin J Verstrepen (Reviewer #3).

The reviewers have discussed the reviews with one another and the Reviewing Editor has drafted this decision to help you prepare a revised submission.

Summary:

The authors evolve yeast in nitrogen-limited conditions. They find that cells increase expression of MEP2, the ammonium permease either by increasing the gene copy number (reported before) or through mutations in the MEP2 transcription factor GAT1. Surprisingly, these mutations reduce the binding of Gat1 to the MEP2 promoter, but still increase the expression of MEP2 by decreasing the expression of its repressor DAL80. This is indeed an interesting and surprising discovery that shows how the connectivity of molecular circuits could impact on their ability to evolve.

Essential revisions:

1) Report expression and fitness for all lineages, as suggested by reviewer #3.

2) Change the model to include auto-regulation of GAT1, as requested below.

3) Discuss the ability to distinguish the contribution of DAL80 vs. GAT1 to the increased MEP2 expression.

4) Discuss the generality of the results.

*Reviewer #1:*

Gresham and colleagues examine the molecular basis of adaptive evolution in an experimental evolution setting. They evolve asexual yeast strains in nitrogen-limited conditions and use whole-genome population sequencing to identify the molecular changes associated with fitness gains. The major target of selection in this context is the expression level of MEP2, coding for an ammonium permease. One of the most advantageous changes at this locus is increase in copy number, which was dissected in previous papers. Other changes that occur are non-synonymous changes in GAT1, a transcription factor positively regulating MEP2. Surprisingly, through functional analyses, the authors show that these amino acid changes reduce the binding of Gat1 to the MEP2 promoter. The decreased binging affinity would contribute to increase the expression of MEP2 by decreasing the expression of DAL80, a repressor of MEP2.

This paper appears to be an important contribution because the evolution of gene expression is generally reduced to a very low level of complexity (including in my own work) by characterizing changes as being in cis or trans and as positive or negative. However, transcriptional networks are very complex and network motifs can produce expression levels and patterns that are not necessarily changing in a simple way, at least not in a way that would be expected based for instance on simple changes in TF affinities. The findings presented here are a good example of this and also illustrate the need to consider molecular changes in the context of molecular networks in order to be able to map fitness changes to phenotypes and genotypes. The experiments are well done, the paper is well written and clearly of interest to a large community.

My major comments would be regarding the interpretation of the data:

- It would be useful to eliminate effects that could come from outside the GAT1-DAL80-MEP2 motif and that could explain why MEP2 expression goes up when Gat1 affinity decreases. The gene expression data produced could be used for this purpose, for instance by showing that other potential MEP2 regulators are not affected in the Gat1 mutant backgrounds.

- Third paragraph of Discussion. It would be useful to have a stronger conclusion as to how the increased expression of MEP2 is achieved. The dynamic model constructed maybe useful in terms of supporting one mechanism or the other, for instance self-regulation of Gat1 versus weaker effect of GAT1 mutations on DAL80 than on MEP2.

*Reviewer #2:*

In this paper Hong and colleagues attempt to understand the functional significance of specific adaptive mutations that evolve in yeast under N_2_ limited chemostat growth. They find that a large number of early adaptive mutations – although not clear what proportion – are missense mutations in one particular TF (GAT1) that acts on multiple genes but in particular is a positive regulator of the N_2_ transporter. These mutations eventually get outcompeted by the expansion of the transporter itself (MEP2) but the early adaptation is (apparently) dominated by the GAT1 missense mutations.

Long story short their argument is that most of the GAT1 are loss (reduction) of function in terms of the binding affinity to the GAT1 binding site but because the downstream targets are regulated through an incoherent FF loop this reduction of binding leads to an *increase* in expression of MEP2 because the GAT1 mutations decrease the expression of the repressor of MEP2, the TF called DAL80.

I was very much prepared to like the paper but was left underwhelmed. The evidence that their model is correct in broad strokes is quite convincing. But they do not seem to be able to link the TF binding to fitness despite having a large number of mutants and do not in any way describe the difference between the two GAT1 mutants they study in great detail. One leads to a much stronger reduction of binding – what does that imply for function?

I am also unclear about the overall message. What do they mean by claiming that the incoherent FFL enhances the ability of yeast to adapt? Is it that in this context loss of function mutants in terms of binding that are more common can generate a gain of function phenotype in terms of expression of the key gene? Is this particular to this structure? What about the other properties of this structure – like the ability to generate a pulse of expression? I was left without a clear sense of how generalizable the findings are and what they mean broadly. Without this sense of general importance it is hard for me to see why this paper out to be published in *eLife* rather than in a more particular molecular biology journal.

*Reviewer #3:*

This study shows that evolution in ammonium-limited chemostats repeatedly selects for modulation of the DNA binding affinity of GAT1, one of the transcription factors controlling nitrogen catabolite repression. Due to it being a part of an incoherent type-1 feedforward loop, this alteration in binding affinity results in an increased expression of MEP2, a high-affinity ammonium transporter. This increase ultimately results in an improved fitness in ammonium-limited conditions. As such, network motifs like feedforward loops might facilitate adaptive tuning of gene expression.

This is an elegant study highlighting the importance of network motifs, such as feedforward loops, a nuance often overlooked up until now in the interpretation of experimental evolution data. One of the major strengths of this study is the reproducibility of the experimental evolution, exemplifying that this mechanism of adaptive gene expression tuning might be more general than previously thought. Granted the authors take some of the below-mentioned comments and concerns into account, this study will certainly improve our understanding of the dynamics of evolution.

• Something that is not really explored in this study, is the influence of GAT1 mutations on its own expression. The data suggest that its expression is also increased (even more than MEP2). This fact is never mentioned in the interpretation of the data, but it might be very important. It is also not included in the mathematical model, which seems a bit simple. As previously said, the expression of GAT1 also depends on the level of GAT1 itself and even on the expression of DAL80. On the other hand, DAL80 supposedly also influences its own expression, and this is also not included in the model. These are very important parts of the feedforward loop, so in my opinion a better model could be made, more resembling the real structure of a feedforward loop. These facts should also be incorporated in the interpretations and discussions.

• The authors suggest several times that the mutation of the DNA-binding domain of GAT1 is a quick way to increase the expression of MEP2 during evolution, before the expression can be increased even more by duplicating MEP2. However, MEP2 expression is never measured in lineages with MEP2 CNVs. As such, the proposed progression of MEP2 expression during evolution is never really shown. As this is one of the main underlying assumptions (increased MEP2 expression equals increased fitness, and CNV has the highest expression), this is a crucial experiment to do. Measure expression and fitness for all lineages, including those with CNVs. Then, correlate this expression with fitness.

---

## [Author Response]

Essential revisions:1) Report expression and fitness for all lineages, as suggested by reviewer #3.

We have now quantified fitness and expression of lineages containing MEP2 amplification alleles. We show that these lineages have increased fitness relative to those containing GAT1 variants, consistent with the observation that they ultimately outcompete the lineages containing GAT1 variants in our evolution experiments. These new results are presented in Figure 2—figure supplement 1B.

2) Change the model to include auto-regulation of GAT1, as requested below.

We studied the effect of GAT1 autoregulation on our model. Using RNAseq, we observe increased expression of GAT1 in strains containing GAT1 variants. Using our model, we find that increases in GAT1 expression linearly amplify expression of MEP2 and thereby contribute to the increased expression of MEP2. These results are now presented in Figure 4—figure supplement 2.

3) Discuss the ability to distinguish the contribution of DAL80 vs. GAT1 to the increased MEP2 expression.

There are four NCR GATA factors: GAT1, DAL80, GLN3 and GZF3. We have added new experiments testing the effect of GLN3 and now more clearly describe the role of each factor and our ability to distinguish their effects. These results are presented in a new Figure 3—figure supplement 2.

4) Discuss the generality of the results.

We have expanded our Discussion regarding the generality of our results.

Reviewer #1:[…] My major comments would be regarding the interpretation of the data:- It would be useful to eliminate effects that could come from outside the GAT1-DAL80-MEP2 motif and that could explain why MEP2 expression goes up when Gat1 affinity decreases. The gene expression data produced could be used for this purpose, for instance by showing that other potential MEP2 regulators are not affected in the Gat1 mutant backgrounds.

This is a good point. The other direct activator of *MEP2* is GLN3 (Figure 1—figure supplement 1). We find that GLN3 expression is not significantly altered in strains containing GAT1 variants (Supplementary file 1B). This is consistent with the fact that the GLN3 promoter does not contain the GATAA binding sites and thus is not regulated by GAT1 or the repressing GATA factors, DAL80 and GZF3. This suggests that the increased expression is not attributable to changes in the activity of GLN3. In addition, our results indicate that GZF3 is not altered in expression making it unlikely that it contributes to changes in MEP2 expression. We have added this point in the main manuscript as follows:

“By contrast, the expression of *GLN3* and *GZF3* is unchanged in the presence of *GAT1* variants (Supplementary file 1B) suggesting that the altered expression of *MEP2* and *DAL80* is attributable to changes in GAT1 binding affinity.”

- Third paragraph of Discussion. It would be useful to have a stronger conclusion as to how the increased expression of MEP2 is achieved. The dynamic model constructed maybe useful in terms of supporting one mechanism or the other, for instance self-regulation of Gat1 versus weaker effect of GAT1 mutations on DAL80 than on MEP2.

We agree. We have now edited this entire paragraph to simplify it and strengthen our conclusion by emphasizing the counter-‐‑intuitive finding that decreased transcription factor binding affinities of an activating TF are consistent with increased expression output:

“The transcriptional output from a promoter is a function of the strength of the promoter, which depends on the identity and number of cis regulatory elements, and the affinity of a transcription factor for those elements. […] Thus, quantitatively different effects of transcription factor variants on promoters of differing strengths, in the context of gene regulatory architectures, can lead to counterintuitive impacts on gene expression.”Reviewer #2:[…] I was very much prepared to like the paper but was left underwhelmed. The evidence that their model is correct in broad strokes is quite convincing. But they do not seem to be able to link the TF binding to fitness despite having a large number of mutants and do not in any way describe the difference between the two GAT1 mutants they study in great detail. One leads to a much stronger reduction of binding – what does that imply for function?

We thank the referee for raising this point. We revisited our data and made some interesting observations. First, we find that the residue mutated in the *GAT1-1* (W321L) allele is only hit once in our repeated evolution experiments whereas the residue mutated in the *GAT1-3* (R345G) allele is the most frequently mutated site (Figure 1E). In addition, *GAT1-1* shows significantly lower fitness compared to *GAT1-3* (R345G) (Figure 2B). Given these results, we suggest that selected GAT1 variants differ in their fitness effects, such that strongly reduced TF binding results in lower fitness increase than does a more moderate reduction in TF binding. We have added this point in the Results section as follows:

“Interestingly, *GAT1-1* (W321L) shows a much stronger reduction in DNA binding affinity compared to *GAT1-3* (R345G) for both tested promoter sequences (Figure 4B and 4C). […] These observations suggest that intermediate reductions in GAT1 binding affinity may be more beneficial than strongly reduced binding. Consistent with this claim, complete loss of GAT1 results in decreased MEP2 expression (Figure 3B) and is deleterious (Figure 2B).”

I am also unclear about the overall message. What do they mean by claiming that the incoherent FFL enhances the ability of yeast to adapt? Is it that in this context loss of function mutants in terms of binding that are more common can generate a gain of function phenotype in terms of expression of the key gene? Is this particular to this structure? What about the other properties of this structure – like the ability to generate a pulse of expression? I was left without a clear sense of how generalizable the findings are and what they mean broadly. Without this sense of general importance it is hard for me to see why this paper out to be published in eLife rather than in a more particular molecular biology journal.

The overall message is that the evolution of gene expression occurs within the context of gene regulatory networks. To study and interpret gene expression evolution, it is essential to consider the properties of the gene regulatory networks as the net effect of altered transcription factor affinities can be unintuitive. We now emphasize point this in the Abstract:

“Our results show that I1-FFLs, one of the most commonly occurring network motifs in transcriptional networks, can facilitate adaptive tuning of gene expression through modulation of transcription factor binding affinities. Our findings highlight the importance of accounting for gene regulatory architectures to understand gene expression evolution.”

And reiterate this point in the Discussion:

“Thus, quantitatively different effects of transcription factor variants on promoters of differing strengths, in the context of gene regulatory architectures, can lead to counterintuitive impacts on gene expression.”

Reviewer #3:[…] Granted the authors take some of the below-mentioned comments and concerns into account, this study will certainly improve our understanding of the dynamics of evolution.• Something that is not really explored in this study, is the influence of GAT1 mutations on its own expression. The data suggest that its expression is also increased (even more than MEP2). This fact is never mentioned in the interpretation of the data, but it might be very important. It is also not included in the mathematical model, which seems a bit simple. As previously said, the expression of GAT1 also depends on the level of GAT1 itself and even on the expression of DAL80. On the other hand, DAL80 supposedly also influences its own expression, and this is also not included in the model. These are very important parts of the feedforward loop, so in my opinion a better model could be made, more resembling the real structure of a feedforward loop. These facts should also be incorporated in the interpretations and discussions.

This is a good point. We explored using a more complicated model including GAT1 autoregulation and DAL80 self-inhibition. However, the model quickly becomes unwieldy because there are too many unknown parameters. Although the model we present is simple, our goal in constructing it was to test whether our experimental observation that decreased affinity for both MEP2 and DAL80 promoters is consistent with increased expression output of MEP2 as this seemed entirely counterintuitive. As we report, the model is consistent with our findings. We did not measure affinity of GAT1 variants for its own promoter, but as the reviewer points out our gene expression data indicate that GAT1 expression is increased. Therefore, we tested the effect of increased GAT1 expression on MEP2 expression output across the entire parameter space studied in Figure 4F. We find that increased GAT1 expression results in a linear increase in MEP2 expression. We have included this result in a new supplementary figure, Figure 4—figure supplement 2

“Figure 4—figure supplement 2.Increasing GAT1 expression results in increased MEP2 expression. We studied the effect of increased GAT1 expression across the same parameter space investigated in Figure 4F. a,We find that when GAT1 expression is increased by a factor of two, the effect of differential activation of *MEP2* and *DAL80* promoters on *MEP2* expression is qualitatively identical and b,for each parameter combination, expression of MEP2 is increased twofold.”

and explain this result:

“Using RNA-seq, we also find that the expression of GAT1 is increased as a result of GAT1 variants (Figure 3A). […] The increase in GAT1 expression likely also contributes to increased MEP2 expression, as modeling an increase in GAT1 expression results in a linear increase in MEP2 expression (Figure 4—figure supplement 2).”

• The authors suggest several times that the mutation of the DNA-binding domain of GAT1 is a quick way to increase the expression of MEP2 during evolution, before the expression can be increased even more by duplicating MEP2. However, MEP2 expression is never measured in lineages with MEP2 CNVs. As such, the proposed progression of MEP2 expression during evolution is never really shown. As this is one of the main underlying assumptions (increased MEP2 expression equals increased fitness, and CNV has the highest expression), this is a crucial experiment to do. Measure expression and fitness for all lineages, including those with CNVs. Then, correlate this expression with fitness.

We thank the referee for the suggestion and have undertaken additional experimentation to address this point. We identified additional lineages from the final time point of each LTEE and quantified DNA and mRNA copy number of *MEP2* as well as the relative fitness of these lineages. We find that CNVs do indeed result in increased expression of MEP2 and that there is a positive correlation between MEP2 expression and fitness in ammonium‑limited conditions. We have added this result in Figure 2—figure supplement 1B and in the text as follows:

“We also isolated evolved lineages with increased copy numbers of MEP2 from each of the three LTEEs and quantified MEP2 copy number, expression level and relative fitness in ammonium-limited conditions (Figure 2—figure supplement 1B). […] The detection of lineages with increased fitness and increased expression of MEP2 resulting from CNVs is consistent with fitness increases in GAT1 variant lineages being attributable to their effect on MEP2 expression.”